# Non-Genetic Diversity in Chemosensing and Chemotactic Behavior

**DOI:** 10.3390/ijms22136960

**Published:** 2021-06-28

**Authors:** Jeremy Philippe Moore, Keita Kamino, Thierry Emonet

**Affiliations:** 1Department of Molecular Cellular and Developmental Biology, Yale University, New Haven, CT 06511, USA; jeremy.moore@yale.edu (J.P.M.); keita.kamino@yale.edu (K.K.); 2Quantitative Biology Institute, Yale University, New Haven, CT 06511, USA; 3Department of Physics, Yale University, New Haven, CT 06511, USA

**Keywords:** bacterial chemotaxis, chemoreceptors, receptor clusters, phenotypic diversity, stochastic gene expression, fluctuations, signal processing, single cell

## Abstract

Non-genetic phenotypic diversity plays a significant role in the chemotactic behavior of bacteria, influencing how populations sense and respond to chemical stimuli. First, we review the molecular mechanisms that generate phenotypic diversity in bacterial chemotaxis. Next, we discuss the functional consequences of phenotypic diversity for the chemosensing and chemotactic performance of single cells and populations. Finally, we discuss mechanisms that modulate the amount of phenotypic diversity in chemosensory parameters in response to changes in the environment.

## 1. Introduction

Bacteria display very diverse morphologies and behaviors. This diversity allows genetically distinct individuals to serve different roles in a community. Within species also, genetic diversity can be an effective strategy to better utilize the environment [1], or evade antibiotic treatment, as is the case with some cancers and bacterial infections [2]. Interestingly, however, substantial phenotypic diversity occurs within populations of bacteria that share the same genome [3,4,5,6]. What is the molecular origin and functional consequences of such non-genetic phenotypic diversity?

Bacterial chemotaxis provides an effective quantitative framework to study non-genetic phenotypic diversity. In their seminal 1976 paper, Spudich and Koshland used the innovative “temporal gradient apparatus” [7], which combined a rapid mixing device with stroboscopic imaging, to characterize the swimming behavior of individual *Salmonella* bacteria in response to pulses of attractant [8]. Surprisingly, behavioral differences between individuals were greater than could be explained by potential mutations accumulating during growth. This and other early work in *Escherichia coli* [9,10] raised questions about cell-to-cell variability in chemotactic behavior and how collective migration occurs in the midst of phenotypic diversity. Since then, non-genetic diversity has been studied extensively in many biological systems, revealing fundamental mechanisms that generate it, including stochastic gene expression and uneven partitioning of biomolecules at cell division [11,12,13,14]. For more general information on the molecular mechanisms and functional consequences of cell-to-cell variability, we refer the reader to these reviews [3,4,13,15,16].

Here, we focus on non-genetic diversity in chemical sensing and the well-characterized chemotaxis system of *E. coli* for which there is a relatively good understanding of the molecular mechanism by which the chemotactic signaling pathway uses external signals to bias cell’s run-and-tumble motility toward favorable locations [17,18]. In this two-component system, five types of chemoreceptors (Tar, Tsr, Trg, Tap, and Aer) form homodimers arranged in trimers of dimers, themselves arranged in hexagonal latices comprising thousands of receptors connected by the scaffold protein CheW and the histidine kinase CheA [19,20,21]. The receptor-kinase complexes are highly cooperative at the level of the CheA activity [22], enabling signal amplification. Upon binding an attractant, receptor-kinase complexes switch to an inactive conformation, reducing the rate of phosphorylation of the response regulator CheY. Due to the phosphatase activity of CheZ, phosphorylated CheY-p concentration rapidly (<1 s) decreases [18], leading to a reduction in the probability to tumble. The prestimulus level of tumbling frequency is later reestablished via the action of CheR and CheB, which methylates and demethylates the cytoplasmic tail of the receptors when they are in the inactive and active conformations, respectively.

The general features of chemotaxis [17,18,20,23] and its quantitative treatment [18,24,25] have been reviewed elsewhere recently. Here, our intent is to review the current understanding of non-genetic phenotypic diversity in *E. coli* chemotaxis and how knowledge of phenotypic diversity affects our general understanding of chemosensing and chemotactic behavior.

## 2. Molecular Mechanisms Underlying Phenotypic Diversity in *E. coli* Chemotaxis

Quantifying phenotypic diversity first requires quantifying phenotypes, which can be accomplished by measuring functional and behavioral parameters in individual cells. For *E. coli* chemotaxis, such parameters include the following: the tumble bias—the fraction of time the cell spends tumbling; the switching frequency—the number of switches per unit of time between the run and tumble states; the pathway gain, or magnitude of the response relative to a stimulus change both at the level of the kinase and of the motor; the adaptation time, or time to return to the original behavioral pattern after a stimulus; and the rotational diffusion, all of which affect chemotactic performance [10,18,25,26,27,28,29,30]. Alternatively, the phenotype of an individual cell can also be quantified without making assumptions about the set of possible behavior states a bacterium can be in (e.g., run-and-tumble). In one approach, each individual cell trajectory was represented as a scatter plot of time points in the space of translational and rotational velocity [31]. Viewed as a two-dimensional probability distribution, the scatter plot defines the motility states the bacterium exhibit over time and the fraction of time it spends in each [32,33].

An important consideration is that phenotypic parameters may vary on multiple time-scales. For example, a cell might exhibit the same tumble bias over its entire life span, or its tumble bias might change from the moment it is born until it divides. At any given instant, multiple stochastic processes operate on different time scales and contribute to phenotypic variability (Figure 1). Below, we review recent advances in current understanding of these various mechanisms and how they affect *E. coli* chemotaxis.

### 2.1. Variation Arising at Cell Division

One mechanism likely to contribute to long-lasting differences between cells is partitioning noise—the unequal distribution of biomolecules between daughter cells during cell division (Figure 1a). General features of partitioning noise and its quantitative treatment have been subject to review elsewhere [13,37]. For chemosensing and motility, the partitioning of organelles, such as molecular motors and large receptor clusters, is likely to contribute to phenotypic variability. However, so far, only few studies have examined this aspect of phenotypic variability.

Recent studies have determined the location and size distributions of chemoreceptor clusters in growing cells [38,39]. At early growth stages, cells contain multiple small receptor clusters, which are positioned along the length of the cell (Figure 1d) [34]. However, as the culture approaches stationary phase, individual cells typically contain only one or two polar clusters. The resulting cell-to-cell differences in cluster size and number are likely to translate into differences in sensitivity to chemical signals. While a direct link between cluster positioning and portioning noise has not been demonstrated directly, an uneven distribution of chemoreceptor complexes could contribute to long-lasting diversity, since some individuals will inherit multiple clusters, while others will inherit none.

### 2.2. Stochastic Pulses of Motility Gene Expression

By far, the most-studied source of phenotypic variability is stochastic gene expression (Figure 1b), where it is understood that transcription and translation events occur in bursts with variable frequency and size, leading to broad distributions of protein expression in isogenic cell populations [6,11,12,14]. Different regulatory schemes can lead to fluctuations in gene expression with various magnitudes and timescales [40,41]. In *E. coli*, the arrangement of chemotaxis genes on multicistronic operons has been shown to buffer the effect of some of these fluctuations in the signaling output by ensuring gene products that participate in the same step of signal transduction are kept at an acceptable ratio [42]. Furthermore, a temperature-sensitive secondary RNA structure upstream of *cheR* was suggested to compensate for temperature-induced differences in the rate of adaptation of cells to signals [43,44]. While these compensatory mechanisms ensure that individual phenotypes remain functional, gene expression noise gives rise to large cell-to-cell variability within this range of functional phenotypic space [45]. For example, the expression ratio of different receptor species varies greatly [46], leading to variation in the sensitivity for different chemoeffectors [36,47].

Recently, a study by the Cluzel lab revealed how regulated stochasticity in gene expression plays an important role in the transcription network governing *E. coli* chemotaxis [35]. In *E. coli*, there are 14 operons collectively encoding the flagellar components and the chemotaxis machinery. These operons contain over 50 genes that are regulated in a three-tier hierarchy. At the top is the class I operon, *flhDC,* which encodes the master regulator of flagella and chemotaxis. Then, FlhDC activates the class II genes encoding the basal body and flagellar hook, as well as the alternative sigma factor FliA, which subsequently activates the class III genes encoding the flagellar filament and chemotaxis network [48].

Using a microfluidic device called the “mother machine” where a single cell’s lineage can be monitored over multiple division events, the Cluzel lab quantified the expression of class I, II, and III genes over multiple generations using fluorescent reporters [35]. While the class I genes were expressed constitutively, class II and III gene expression occurred in pulses lasting multiple generations interspersed with long periods of inactivity (Figure 1e). Interestingly, these pulses did not depend on the regulation of *flhDC* transcription but were instead shown to require interactions between FlhDC and its post-translational regulator YdiV, which amplified transcriptional noise, allowing small temporal fluctuations in *flhDC* transcription to generate large pulses of class II expression. In short, transcriptional noise, coupled with a post-translational circuit, converts constitutive expression of class I genes into pulsatile expression of class II and III genes.

The pulsatile expression pattern of class II genes was further explored in theoretical work that proposed a possible molecular mechanism [49]. Simple stochastic models of class I and class II gene expression successfully capture the expected distributions of class II expression. By fitting these models to single-cell data [35] of class II expression with and without YdiV, Sassi et al. found that YdiV’s functions were twofold: first, it makes class II gene expression ultrasensitive to changes in FlhDC concentration. Second, it allows the system to integrate FlhDC expression over time to filter away small fluctuations. As such, they proposed a molecular mechanism where YdiV sequesters FlhDC, such that small changes in FlhDC expression lead to ultrasensitive increases in class II promoter activity [49]. This simple sequestration mechanism allows the expression of motility genes to follow a filter-and-integrate method of pulse generation that is qualitatively distinct from other known biological pulse generators that rely on bistability in transcriptional feedback loops [50,51].

What is the end result of such an expression pattern? Ultimately, stochastic gene expression generates a wide diversity of sensory and motile phenotypes that may contribute to phenotypic changes over the course of cell growth as observed in behavioral studies [33]. An interesting hypothesis is that cells in generations following a valley of little expression could have different chemotactic phenotypes than those following a pulse of expression. Such an expression pattern could also play a role in how individual cells within one population sort themselves in space during collective migration, as described in later sections.

### 2.3. Spontaneous Temporal Fluctuations in Pathway Activity

In addition to partitioning and gene expression noise, the behavior of individual cells is also affected by temporal fluctuations that arise due to the inherent stochasticity of the chemical reactions of the chemotaxis signal transduction pathway (Figure 1c). The first study to characterize these fluctuations measured the rotational direction of latex beads attached to flagella in cells that were adapted to a motility buffer devoid of any signal [52]. Analyzing the power spectra of these time-series revealed surprisingly large fluctuations in tumble bias with characteristic times scales on the order of 10s of seconds. Blocking adaptation by receptor methylation eliminated these fluctuations, while titrating the amount of methyltransferase CheR changed their time scale and amplitude [52].

These data can be recapitulated with a coarse-grained model of the signaling pathway in which the rates of methylations and demethylations follow Michaelis–Menten kinetics with a sub-linear dependency on the activity of the receptors [45,52,53,54,55]. This model assumes that in vivo, CheR and CheB work near saturation [53,54], which is consistent with population average FRET measurements of adaption kinetics [56] and with stoichiometric measurements—CheRB are expressed orders of magnitude less than the receptors [57]. A consequence of the saturated enzyme kinetics hypothesis is that the steady-state activity of the system becomes ultra-sensitive [58] to the ratio of CheR and CheB abundances, which can contribute to the amplification of spontaneous fluctuations in the activity of the receptors [45,52,54,55].

Recently, the wide availability of high-sensitivity cameras allowed the classic CheZ/CheY FRET method [59]—originally developed to measure kinase activity in a population of cells—to be adapted for quantifying kinase activity in single cells [36,60,61,62] (Figure 1f). Measurements of the kinase activity in individual unstimulated cells revealed large fluctuations in the kinase activity that could be attributed to two different sources. The first source was traced back to the activity of CheR and CheB. Similar to previous results [52], these fluctuations have characteristic time scales of tens of seconds and were strongly suppressed in *cheRB^-^*. Moreover, perturbations to the CheB phosphorylation feedback loop caused changes in steady-state kinase activity that were consistent with an ultrasensitive dependency of the activity of receptor-kinase complexes on the ratio of CheR/CheB abundances, and the role of the CheB phosphorylation feedback loop in reducing such ultra-sensitivity [36,54].

Interestingly, the magnitude of the fluctuations measured at the kinase were larger than previous estimates from fluctuations measured at the motor output [36], suggesting that a second source of fluctuations might be present. Indeed, measurements in *cheRB^-^* cells stimulated with a constant sub-saturating stimulus revealed methylation-independent fluctuations with time scales of 100s of seconds [36,60]. These fluctuations were eliminated when the scaffolding protein that mediates allosteric interactions in the receptor-kinase complexes, CheW, was replaced with the mutant CheW-X2 [60], which is a mutant that abolishes allosteric interactions but maintains signaling capabilities [22,63]. These large fluctuations also depended on the composition of receptor clusters and became almost switch-like between a few discrete states in cells that only expressed Tsr [36]. Together, these data suggest that the second source of fluctuations is caused by thermal fluctuations in highly cooperative receptor clusters [36,60].

While a full biophysical model is still missing, the emerging picture is that receptor clustering and adaptation kinetics both contribute to temporal fluctuations in the chemotaxis pathway. While thermal fluctuations and allosteric interactions in receptor clusters can introduce large fluctuations over very long timescales (up to 100s of seconds), the stochasticity of methylation events contributes smaller fluctuations on time scales of 10s of seconds. Interestingly, receptor clustering is required for fluctuations to be observed even in the presence of methylation and demethylation, which is probably because of the role of “assistance neighborhoods”—in which CheR (or CheB) tether to one receptor and methylate (or demethylate neighboring receptors—in signal amplification and adaptation kinetics; see [60,63,64,65,66]. The ultra-sensitivity of the kinase activity with respect to the ratio of CheR/CheB is also a source of phenotypic variability, which becomes more apparent when the buffering effect of the CheB phosphorylation feedback is removed [36].

## 3. Functional Consequences of Phenotypic Diversity on Chemosensing and Chemotactic Performance

Phenotypic diversity can be beneficial for populations of bacteria, especially in the face of environmental uncertainty [3,4,5,67]. In this section, we review how phenotypic diversity impacts chemotactic performance, how fluctuations in the chemotactic signaling pathway affect a cell’s information-processing ability, and how cell populations perform collective behaviors despite containing a multitude of sensory and behavioral phenotypes.

### 3.1. Functional Consequences of Phenotypic Diversity

Populations of *E. coli* display widely diverse swimming behaviors. Some of the first studies of run-and-tumble motion in *E. coli* demonstrated that adaptation time [8], tumble bias, and swimming speed [10] vary from cell to cell. New methodologies in single-cell tracking have expanded on these measurements, allowing precise, high-throughput quantification of single-cell behavioral parameters including the run-speed, turning angle, and tumble bias [26,27,29,30,31]. Meanwhile, optical tweezers have provided high time-resolution measurements of adaptation in single cells [68,69].

How do the processes described in previous sections contribute to this diversity? Variable expression of flagellar components, chemoreceptors, and sensory pathway components may all be relevant for certain aspects of *E. coli*’s behavioral diversity. For example, variation in the CheR/CheB ratio leads to a variation in tumble bias. Interestingly, the variance of this distribution depends primarily on CheB expression and not CheR, suggesting that mutations in regulatory elements might allow populations to adapt both the mean and variance of tumble biases by changing CheB and CheR expression [31,45]. Variations in tumble bias has profound consequences on chemotactic performance. By tracking individual cells in a static gradient of attractant, Waite et al. measured the relationship between tumble bias and drift velocity of individual cells [70]. They found that in liquid, wild-type cells with low tumble bias (TB = 0.01) climb gradients much faster than cells with intermediate tumble bias (TB = 0.2), as predicted by theory [71]. Next, by titrating the level of expression of CheR (or CheY) in the population, they could establish a causal relationship between CheR (or CheZ), tumble bias, and gradient climbing speed. Additionally, variation in pathway gain has been inferred from both single-cell tracking [28] and a microfluidic T-maze that sorts the population by chemotactic drift speed [72]. Diversity in pathway gain may result from diversity in the expression of different receptor species [46].

One way in which the diversity of sensory and swimming phenotypes benefits bacteria is to increase the overall chemotactic performance of the population beyond that of the mean phenotype within the population. This was apparent in Waite et al.’s experiment mentioned above. At approximately wild-type CheR expression, the motion of the population’s mean position closely followed the position of cells with average tumble bias. However, at lower average CheR expression, the population’s mean position ascended the gradient more quickly than cells with average tumble bias. This discrepancy between population performance and the performance of the mean phenotype can be explained by Jensen’s inequality [73]: for a convex function of a random variable f(x), the average of value of the function with respect to the random variable x, E[f(x)], is equal or greater than the value of the function evaluated at the average x, f(E[x]). Since the relationship between phenotype and gradient-climbind speed is nonlinear at low CheR expression and linear at high CheR expression, populations in the low expression regime climb gradients more quickly on average when there is variation in tumble bias [70].

In addition to directly modulating population performance, phenotypic diversity can allow populations to survive in diverse, fluctuating environments [3,5]. Bet-hedging is a general ecological strategy where populations generate individuals with phenotypes poorly adapted for the current environment but well adapted for a different environment. This strategy ensures the population’s survival after an environmental change [74,75]. Theoretical work in *E. coli* chemotaxis demonstrated that variability in tumble bias and adaptation time allows the population to navigate toward nutrients appearing at different distances away from the population [45]. Since a single phenotype cannot optimally locate both near and far attractant sources [76], generating diverse phenotypes is necessary for the population’s survival in the wild, where nutrients may appear any distance from the population.

An attractive hypothesis is that variation in behavioral parameters also allows *E. coli* to perform well in both liquid and porous solid environments, where the optimal phenotypes are likely different. New methods developed in the Datta lab are revealing how *E. coli* navigate disordered porous environments. Three-dimensional porous environments were constructed using small, clear, hydrogel particles, allowing for imaging of single bacteria in environments of different average pore diameter [77,78]. In such environments, cells do not employ a run-and-tumble strategy, as in unconfined liquid environments. Instead, they hop between openings in the media after changes in flagellar bundling allow them to escape confined spaces [77]. Efficient navigation in such an environment could require a different tumble bias than in liquid media; as such, it is possible that phenotypic diversity allows the population to bet-hedge against the physical properties, as well as the chemical makeup, of the environment.

### 3.2. Consequences of Temporal Variation in the Chemosensory Pathway

As discussed above, behavioral variation within populations can be beneficial; are signaling fluctuations within individual cells similarly beneficial? Such fluctuations corrupt the transmitted signal [79]. Since the system consumes energy (in the form of ATP for CheY phosphorylation and S-adenosyl methionine for methylation), it should in principle be possible to decrease the magnitude of fluctuations, as shown in a recent theoretical analysis [80]. Why then has the system evolved to maintain these fluctuations?

One reason may be the multi-task nature of the chemotaxis signaling pathway: it must process external signals to bias chemotactic behavior in the presence of attractant gradients, but it must also drive swimming behavior in the absence of gradients to find better conditions for growth. Since the first observation of signaling fluctuations [52], it has been suggested that such fluctuations lead to fat-tailed distributions of run lengths over a few decades [54,81]. A random walk with such a distribution of run-lengths approximates what is called a ‘Levy flight/walk’. In a Levy walk, long runs occasionally bring the cell to new, unexplored territory, leading to more efficient exploration of the environment [82]. Numerical simulations have also shown that Levy-walk behavior helps cells respond to and climb shallow chemoattractant gradients more efficiently by increasing the coordination between individual flagellar motors and enabling cells to detect very small signals by integrating stimuli over long durations [83,84].

Despite intensive theoretical study of the causal relationship between signaling fluctuations and Levy walks of free swimming cells [81,82], experimental evidence that *E. coli* performs a Levy walk has remained circumstantial. Recently, however, three-dimensional tracking of individual bacteria over a long time have revealed how in the absence of chemoattractant, signaling fluctuations drive long tails in the distribution of run durations [26]. Comparing wild-type behavior with that of a mutant strain lacking signaling noise indicates that a consequence of the long tails in run duration is a superdiffusive swimming behavior consistent, at least over a decade and a half, with Levy walk [85] (Figure 2a). Levy flight has also been observed in swarming bacteria [86]. Thus, fluctuations in the chemotaxis pathway can modulate behavior for efficient spatial exploration over multiple length scales.

The multi-task nature of the chemotaxis signaling network raises the question of whether the system still effectively operates as an information-processing system in biasing run-and-tumble behavior. Combining theory and quantitative measurements, Mattingly and Kamino et al. measured information-theoretic properties of the *E. coli* chemotaxis pathway for cells climbing shallow gradients [62]. Due to the presence of the signaling fluctuations, the amount of information *E. coli* acquired from the environment during chemotaxis was quite low—on the order of 0.01 bits/s in a shallow gradient where chemoattractant concentration varies on centimeter scale. This is far less than the 1 bits/s the cell needs to determine within a 1 s run whether it is swimming up or down of the gradient. However, the same experiments revealed that *E. coli* used that little information efficiently, achieving drift speed up the gradient around 70% of the theoretical maximum achievable performance. Thus, in spite of the large signaling noise, the chemotaxis signaling pathway seems to have been selected for the efficient usage of information it gathers from the environment.

### 3.3. Spatial Sorting of Chemotaxis Phenotypes

Despite containing diverse sensory and swimming phenotypes, populations of *E. coli* can perform collective migration, moving as a cohesive wave that chases a front of attractant generated by the cells consuming the environment [9,88]. How can populations achieve collective behavior in the midst of phenotypic diversity?

Recent work that combined theory and experiments conducted in microfluidics revealed that population migrating in an Adler wave form resolves the conflict between individuality and collective behavior by spontaneously organizing the position of the different phenotypes within the traveling wave [87]. It turns out that not all positions in the wave are equally difficult to navigate. Near the front, the gradient is shallower, and therefore, chemotaxis is more difficult. However, further back, navigation is easier, since the gradient is steeper due to a higher density of cells consuming the environment at that position. These position-dependent differences in signal strength cause the more performant cells to locate at the front where the signal is weaker and the lower performers to locate near the back where the gradient is steeper. This spatial organization can be visualized using mixtures of cells that express CheZ at different levels: CheZ controls tumble bias, which in turn affects the speed at which a cell can climb a gradient [70]. The cells with the higher tumble bias (lower level of expression of CheZ) are located behind those with lower tumble bias (red and blue dots in Figure 2b). Moreover, the distance between the peaks of the two populations traveling together increased with the distance in phenotypic space. Thus, phenotypic diversity leads to spatial sorting withing migrating groups of bacteria.

At the very back of the wave, the gradient becomes shallow again, and the concentration of signal eventually goes below detection level. Thus, phenotypes located near the back are at a higher risk of falling behind the migrating front, leading to a slow leakage of cells of the weakest phenotype at the back of the wave. Recent studies of chemotactic populations find that expanding populations are characterized by a group of pioneer cells that explore new environments and leave behind settler offspring that grow [1,89]. That is to say, bacteria may generate both chemotactic and sessile individuals as part of a division of labor strategy [15,90,91,92] to best utilize their environments.

## 4. Diversity Tuning: Modulating the Degree of Phenotypic Diversity in Response to the Environment

Although phenotypic diversity can help a cell population survive in some environments, theoretical work suggests that the optimal degree of phenotypic diversity depends on environmental conditions [93]. Bet-hedging is beneficial in conditions with sufficient uncertainty [75], but in conditions with clear environmental cues, a “tracking strategy”—where phenotypes closely match the environment—is preferred [94]. This raises the question of whether and how cell populations adjust the level of phenotypic diversity in different environmental conditions. Recent studies have identified a few concrete examples where such diversity tuning occurs.

Phenotypic diversity tuning through gene expression changes was found at least in two different contexts. A detailed study of microbial dinitrogen (N_2_) fixation in *Klebsiella oxytoca* has revealed that the level of phenotypic heterogeneity in metabolism is modulated by the limitation of substrates [95]: if N_2_ is the only available substrate, cells commit to fixing N_2_, showing little cell-to-cell variation in metabolism; however, when a small amount of the more favorable substrate NH_4_^+^ is present, the N_2_ fixation rate diversifies, resulting in large cell-to-cell variation in the N_2_ fixation rate. This diversification enables some cells to fully exploit the presence of NH_4_^+^ while maintaining a fraction of cells that can survive NH_4_^+^ depletion.

Another striking example of diversity tuning through gene expression changes was found in the trimethylamine oxide (TMAO) respiratory system in *E. coli* [96,97]. TMAO is an alternative terminal electron acceptor to oxygen and is utilized by the TMAO reduction machinery. In this system, when TMAO is present in an anaerobic condition, cells express TMAO reductase with low variation, fully committing to utilizing TMAO as a terminal electron acceptor; however, when TMAO is present in an aerobic condition, cells express TMAO reductase at the same mean but with higher variability. The variance of the phenotype is adaptively controlled by oxygen-dependent repression of TMAO signaling genes, providing a fitness advantage when oxygen availability rapidly drops [96].

In the above examples, the diversity of gene expression profiles changes in response to the environment. However, changes in the gene expression profile are relatively slow, taking tens of minutes. How could a population respond to environmental changes that occur on a much shorter time scale? A recent study of the *E. coli* chemotaxis pathway discovered an implementation of diversity tuning through a solely post-translational mechanism, which is capable of modulating diversity in tens of seconds [61]. The phenotype investigated was the response sensitivity of the chemotaxis pathway (defined as the inverse of the ligand concentration eliciting half-maximal response). In the absence of ambient chemoattractant, the level of variation in response sensitivity (measured using single cell FRET) was high; however, in the presence of ambient chemoattractant, the level of diversity sharply decreased for both major chemoreceptors Tar and Tsr (Figure 3a,b). The experiments were conducted in an auxotrophic condition, and therefore, no gene expression change is involved in the mechanism of diversity tuning. By extending a two-state receptor model that had been calibrated only by population-averaged data [24,98,99] to include cell-to-cell variation, Kamino et al. showed that the diversity tuning can be explained only by the environment-dependent methylation of chemoreceptors and a standing cell-to-cell variation in their allosteric coupling. Mechanistically, the abrupt transition in the amount of diversity could be traced back to the nonlinearity in the response function of the receptor cluster, which—as shown earlier by Keymer et al—switches between two different response regimes when the free energy difference between active and inactive unbound receptors changes sign due to methylation [98].

What could be the functional consequences of sensory diversity tuning? One possibility is that it enables a population to transition between a bet hedging strategy and a tracking strategy in a background-stimulus dependent manner [61] (Figure 3c). For chemotactic bacteria, the zero-stimulus background is maximally uncertain in that there is no information about the nature and magnitude of future signals, and therefore, the diversification of sensitivity for different ligands could be beneficial in improving the population’s readiness to detect many different signals. On the other hand, once a relevant signal is perceived, large cell-to-cell variability could be harmful, leading to desensitization or sensory saturation depending on whether the sensitivity is too high or too low. Instead, focusing on tracking the signal by reducing variability is beneficial.

Although the sensory diversity decreases after adaptation, there is still significant cell-to-cell variation in response sensitivity. Why are these cells maintained in the population if their low sensitivity, and thus low pathway gain, decreases their performance when climbing chemical gradients? Recent theory work has implied that having cells with low sensitivity is advantageous because it improves the performance of the population in navigating and colonizing environments with many nutrient patches [100]. Cells with higher gain climb gradients quickly but are easily trapped around local maxima. On the other hand, cells with lower gain are slow to climb gradients but more likely to detach from a local maximum and explore new territory. Since the response gain will fluctuate over multiple time scales ranging from that of signaling fluctuation (≈10–100 s) to cell division (≈1 h), *E. coli* chemotaxis with gain variation resembles a stochastic sampling method called simulated tempering known to be efficient in sampling from probability distributions with many local maxima and minima.

## 5. Conclusions and Outlook

In recent years, it has become clear that phenotypic diversity and temporal fluctuations have a functional role in the chemotactic behavior of *E. coli*. Currently, however, we only understand this role from either a theoretical perspective or under controlled laboratory conditions. It remains unclear how phenotypic diversity and temporal fluctuations impact individual and population-level strategies in natural settings.

One question we must answer to understand the role of diversity in natural bacteria is how the environment affects the degree of phenotypic heterogeneity. While the diversity of sensitivities to chemoattractant can be modulated by the presence of background signals [61], whether other parameters of the signal transduction pathway are similarly tuned by the environment requires further study. Furthermore, how might different nutrient environments affect the diversity of chemotaxis gene expression profiles? While the effect of different nutrient environments on average chemotactic behavior is becoming clearer [101], the diversity of chemotactic phenotypes as a function of the environment is still unknown.

Another area that currently lacks understanding is the role of temporal fluctuations in bacteria with different swimming modes. Recent work has focused on the mechanism [36,60] and function [85] of temporal fluctuations in the chemotaxis signaling pathway of *E. coli*, but their existence and function in other bacteria, especially those that do not navigate by runs and tumbles, has yet to be verified. Do all navigational strategies benefit from fluctuations in their signaling pathways, or is the usefulness of such fluctuations unique to run-and-tumble navigation? Does the architecture of a bacteria’s chemotactic signaling pathway [102] or chemoreceptor arrays [21] affect these fluctuations? Answering these and other questions will be important to generalize our understanding of the chemosensory system of *E. coli*.

## Figures and Tables

**Figure 1 ijms-22-06960-f001:**
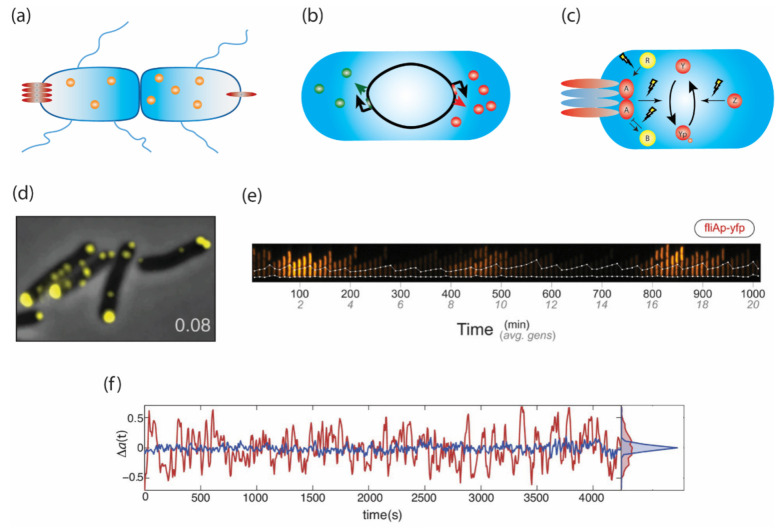
Phenotypic diversity arises from processes operating at different timescales. (**a**) Unequal partitioning of proteins and organelles upon cell division, (**b**) stochastic gene expression, and (**c**) stochasticity in the chemical reactions of signaling pathways all contribute to phenotypic diversity. (**d**) The number and size of chemoreceptor clusters varies from cell to cell due to random partitioning. This variation may lead to diverse sensitivities to stimuli. (**e**) The expression of class II motility genes such as *fliA* occurs in pulses lasting multiple generations. These pulses are stochastic, likely generating cells with very different sensory and swimming capabilities. (**f**) Fluctuations in the kinase activity of the chemotaxis network. In red are cells containing the adaptation enzymes CheR and CheB, while in blue are *cheRB* deletants. The cycle of methylation and demethylation, as well as receptor cooperativity both contribute to fluctuations. Panels d, e, and f are adapted from Koler et al., 2018 [34]; Kim et al., 2020 [35]; and Keegstra et al., 2017 [36].

**Figure 2 ijms-22-06960-f002:**
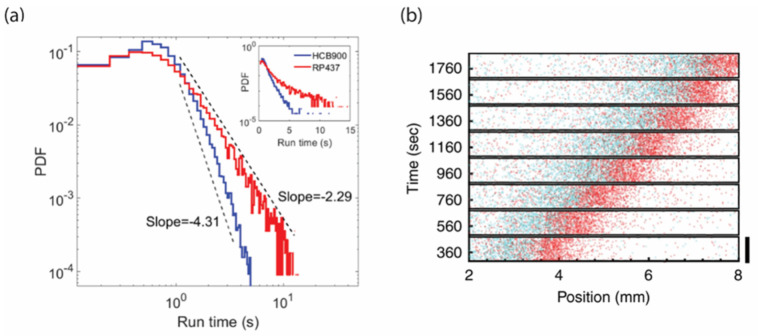
Phenotypic diversity and signaling fluctuations affect ecological performance. (**a**) The distribution of run lengths in wild-type cells (red) has a fatter tail than in cells with the same mean tumble bias but no fluctuations in CheY phosphorylation (blue). This distribution of run lengths allows cells to efficiently explore the environment by Levy walk instead of a pure random walk. (**b**) The conflict between individuality and collective behavior is resolved by spatial sorting of phenotypes in traveling waves of bacteria. Cells with lower tumble bias (red) climb attractant gradients more quickly than cells with higher tumble bias (blue), and therefore are located near the front of the wave. Panels a and b are adapted with permission from Huo et al., 2021 [85]; Fu et al., 2018 [87].

**Figure 3 ijms-22-06960-f003:**
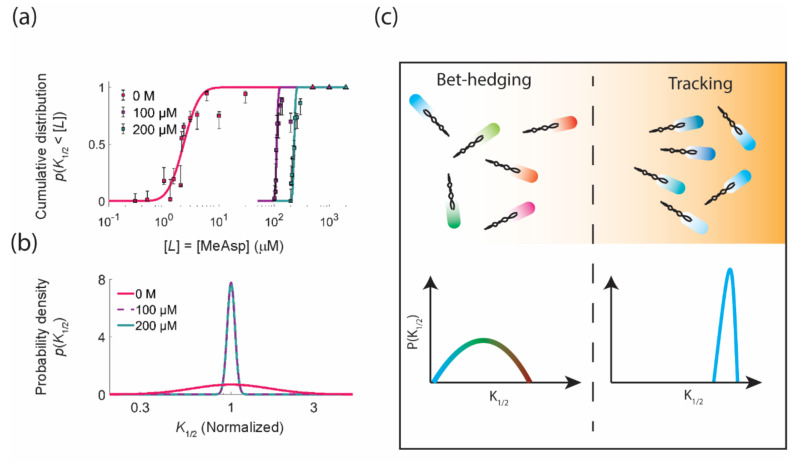
Bacteria tune the degree of sensory diversity in response to chemoattractant. (**a**,**b**) The cumulative distribution and probability density functions of the inverse sensitivity (K_1/2_ defined as the stimulus level that elicits half-maximal response amplitude) for methyl-aspartate (MeAsp). When there is no chemoattractant in the environment, the sensitivity distribution is broad. However, when cells are exposed to non-zero chemoattractant concentrations, the sensitivity distribution sharply narrows. (**c**) The ability to modulate the degree of diversity in the sensitivity distribution according to the background stimulus could allow the population to switch between two navigational strategies in different environments. When no signals are present, cells have no information about future environmental signals, so they diversify their sensitivities for different ligands, improving the readiness for potential upcoming conditions (bet-hedging strategy). However, once a signal is detected, the population is benefited by having most cells being sensitive to the change in the signal level (tracking strategy). Panels a and b are adapted with permission from Kamino et al., 2020 [61].

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
