# Peer review of "Non-Genetic Diversity in Chemosensing and Chemotactic Behavior"

_ijms, 2021, doi:10.3390/ijms22136960_

Round 1

Reviewer 1 Report

Non-genetic Diversity in Chemosensing and Chemotactic Behavior provides a comprehensive overview of the underlying mechanisms for phenotypic diversity in the ability of E. coli to sense and respond to attractants in the environment. The review is timely, current, and very well written.  The work reviewed here does focus on E. coli- and it is most likely beyond the scope is this review to expand into studies in other species.  Therefore, it would be helpful to better indicate that this review does specifically focus on advances in this area in E. coli (and also note where other species are the subject of the study e.g., Klebsiella oxytoca).

Reviewer 2 Report

This is a timely review of the sources and consequence of non-genetic diversity in bacterial chemotaxis. Bacterial chemotaxis is an important model system for the molecular underpinnings of behavior, and has been extensively studied for decades. It is indeed surprising that a similar review on non-genetic diversity has not been written before, to the point that I initially felt certain I read similar reviews many times. But upon going through my own references only the cited 2018 Waite, Frankel, and Emonet paper on more general behavioral variability came close to this scope. The paper is will written, covers the stated topic very well, and contributes to the field by synthesizing many perspectives. Congratulations to the authors, this will certainly be an important review to many people.
